# Linear discriminant analysis reveals hidden patterns in NMR chemical shifts of intrinsically disordered proteins

**Javier A. Romero**[1⊙], **Paulina Putko**[1⊙], **Mateusz Urbańczyk**[2], **Krzysztof Kazimierczuk**[1]*, **Anna Zawadzka-Kazimierczuk**[3]*

**1** Centre of New Technologies, University of Warsaw, Warsaw, Poland, **2** Institute of Physical Chemistry, Polish Academy of Sciences, Warsaw, Poland, **3** Biological and Chemical Research Centre, Faculty of Chemistry, University of Warsaw, Warsaw, Poland

⊙ These authors contributed equally to this work.
* k.kazimierczuk@cent.uw.edu.pl (KK); anzaw@chem.uw.edu.pl (AZK)

**Data Availability Statement:** All relevant data are deposited in Zenodo (10.5281/zenodo.7032142)

## Abstract

NMR spectroscopy is key in the study of intrinsically disordered proteins (IDPs). Yet, even the first step in such an analysis—the assignment of observed resonances to particular nuclei—is often problematic due to low peak dispersion in the spectra of IDPs. We show that the assignment process can be aided by finding "hidden" chemical shift patterns specific to the amino acid residue types. We find such patterns in the training data from the Biological Magnetic Resonance Bank using linear discriminant analysis, and then use them to classify spin systems in an α-synuclein sample prepared by us. We describe two situations in which the procedure can greatly facilitate the analysis of NMR spectra. The first involves the mapping of spin systems chains onto the protein sequence, which is part of the assignment procedure—a prerequisite for any NMR-based protein analysis. In the second, the method supports assignment transfer between similar samples. We conducted experiments to demonstrate these cases, and both times the majority of spin systems could be unambiguously assigned to the correct residue types.

## Author summary

Intrinsically disordered proteins dynamically change their conformation, which allows them to fulfil many biologically significant functions, mostly related to process regulation. Their relation to many civilization diseases makes them essential objects to study. Nuclear magnetic resonance spectroscopy (NMR) is one of the methods for such research, as it provides atomic-scale information on these proteins. However, the first step of the analysis – assignment of experimentally measured NMR chemical shifts to particular atoms of the protein – is more complex than in the case of structured proteins. The methods routinely used for these proteins are no more sufficient. We have developed a method of resolving ambiguities occurring during the assignment process.

In a nutshell, we show that an advanced statistical method known as linear discriminant analysis makes it possible to determine chemical shift patterns specific to different

and GitHub (https://github.com/gugumatz/LDA-for-mapping-IDPs).

**Funding:** JAR, PP and KK thank National Science Centre of Poland (www.ncn.gov.pl) for its support in the form of an OPUS grant (2019/35/B/ST4/01506). The funders had no role in study design, data collection and analysis, decision to publish, or preparation of the manuscript.

**Competing interests:** The authors have declared that no competing interests exist.

types of amino acid residues. It allows assigning the chemical shifts more efficiently, opening the way to a plethora of structural and dynamical information on intrinsically disordered proteins.

This is a *PLOS Computational Biology* Methods paper.

## Introduction

Intrinsically disordered proteins (IDPs) play an essential biological role in eukaryotes, being involved in differentiation, transcription regulation, spermatogenesis, mRNA processing and many other processes [1]. Research into IDPs is thus crucial, but it is also very challenging, as the high mobility of the polypeptide chain prevents crystallization, hampering the use of X-ray crystallography. This dynamic behavior by IDPs also prevents the use of cryogenic electron microscopy [2]. In light of this, nuclear magnetic resonance spectroscopy (NMR) remains the most appropriate method for atomic-level analysis, providing information on structure, dynamics and interactions with other molecules.

The most important observables in NMR are the resonance frequencies of nuclear magnetic moments placed in an external magnetic field. These frequencies, typically expressed in a chemical shift scale, depend on the local moieties of the nuclei. In particular, they are characteristic of the different amino acid residue types in a protein. In the case of folded proteins, the chemical shifts are further influenced by the secondary structure, but for IDPs the effect is far weaker [3]. Although IDPs do not behave in a purely random way and often form "compact states" [4], these states are only adopted transiently. For this reason the structure-induced chemical shift effects are averaged, resulting in spectra that can be very crowded and difficult to analyze.

The assignment of NMR signals to the nuclei of a protein is based on an analysis of a set of heteronuclear ($^1$H, $^{13}$C, $^{15}$N) spectra that provide information about the sequential connectivities of chemical shifts. Such a set can include standard three-dimensional spectra such as HN (CA)CO or HNCA [5, 6], for example, but it can also use those of higher dimensionality (4D and more) to resolve signal overlap [7, 8]. The resonance assignment procedure comprises several steps (see Fig 1). The first step is peak picking, followed by the formation of spin systems. Each spin system contains information on the chemical shifts of nuclei interacting via scalar coupling (typically belonging to two adjacent residues, *i* and *i-1*, although some experiments can reach further [8]). To this end, peaks from different spectra that share certain resonance frequencies must be gathered. Next, by finding identical chemical shifts in different spin systems, sequential connectivities are established and spin-system chains are formed. Although the protein's primary structure is a single, long linear chain of amino acid residues, the analysis of sequential connectivities in NMR spectra almost always leads to the formation of many shorter chains. The chains are interrupted when chemical shifts are found to be missing from the linking spectra, connectivities are ambiguous, or at proline positions (in the case of amide proton-detected experiments). In the particular case of IDPs, many very short chains appear due to poor peak separation.

The final step is mapping the formed spin system chains onto the protein sequence. This mapping typically involves identifying characteristic amino acids by using the chemical shift statistics found in the Biological Magnetic Resonance Data Bank (BMRB), for example [9]. IDP-tailored statistics are also available [10], providing additional information about the influence of adjacent residues on the chemical shifts. The dependence of chemical shifts on

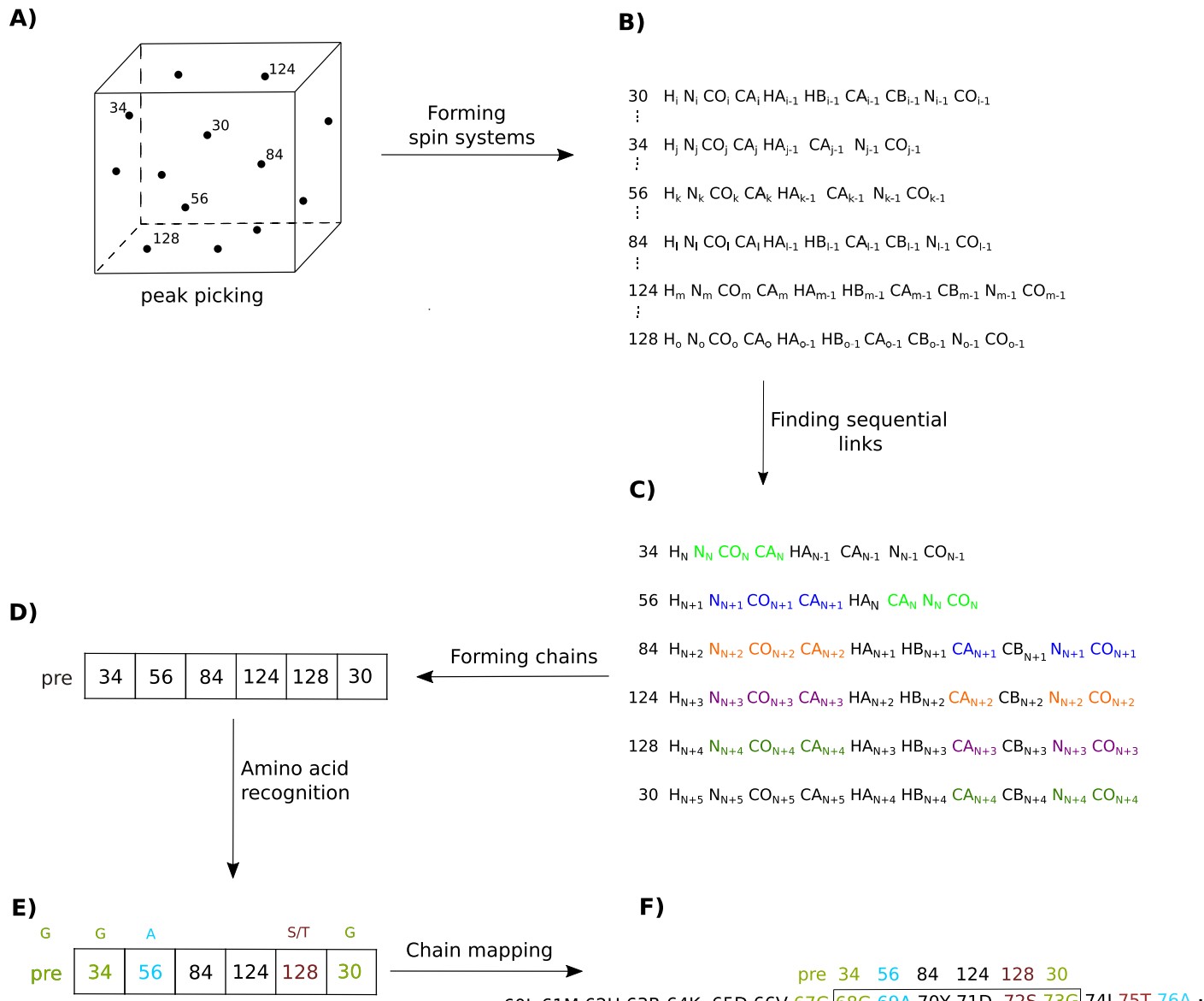

**Fig 1.** The sequential assignment workflow: A) peak picking for all spectra; B) forming spin systems; C) finding sequential links (nuclei whose chemical shifts are used to find the connections between adjacent amino acid residues are marked with different colors); D) forming chains (the numbers of consecutive spin systems are given in boxes; the label "pre" denotes the residue preceding the formed chain for which some chemical shifts are known; E) amino acid recognition based on characteristic chemical shifts; F) chain mapping onto the protein sequence.

neighbors further away (*i*-2, *i*+2), pH, ionic strength and temperature can also be exploited [11]. Typically, $C_\beta$ chemical shifts are used for mapping, with further assistance from $H_\beta$, if available, or $C_\alpha$. Certain amino acid residue types can be excluded based on the structure of the residue [12]. For example, the presence of $H_\beta$ or $C_\beta$ chemical shifts indicates a non-glycine residue. The presence of two different $H_\beta$ chemical shifts excludes alanine, isoleucine, threonine and valine, as these residues do not contain chemically inequivalent $H_\beta$ protons. In short, we can easily recognize alanine, glycine, serine and threonine, but more detailed analysis is needed for other amino acid residue types. Some of the above-mentioned recognition

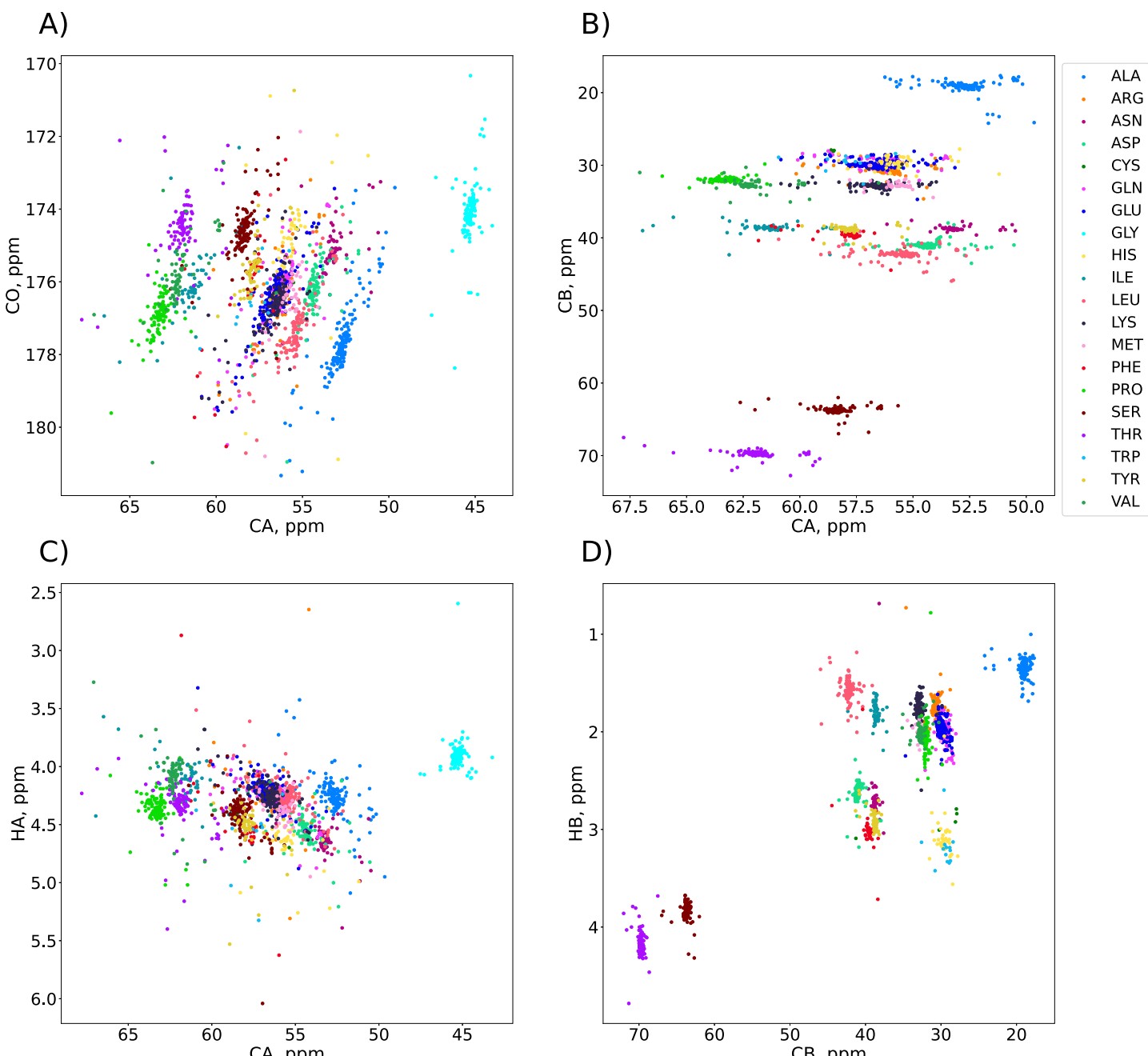

**Fig 2. Chemical shifts of 17 unfolded proteins from the BMRB, marked with colors according to their amino acid residue types: A) Cα/CO plane; B) Cα/Cβ plane; C) Cα/Hα plane; D) Cβ/Hβ plane.** Although the chemical shifts are specific to the residue types, none of the 2D planes provides satisfactory separation allowing unambiguous assignment.

procedures are embedded in automatic assignment programs [13–16]. Clearly, the longer the chain in question, the easier the mapping step. In the case of shorter chains, which are very common in IDP analysis, it is essential to recognize the amino acids of as many residues as possible in order to map the chains accurately.

Fig 2 shows the distributions of the chemical shifts in a set of 17 IDPs from the BMRB. Clearly, in most cases, using just two chemical shifts for amino-acid recognition is insufficient: On two-dimensional planes, regions corresponding to different types of residues often overlap partially or even completely. Although some residue types can be clearly recognized—alanine, glycine, serine and threonine, for instance—recognizing others can be problematic. The $C_\beta$-$H_\beta$ plane turns out to be the best choice for grouping residue types into spectral regions. Nonetheless, even here the regions are not fully separated. Adding more chemical shifts would improve the separation of regions, but also complicate visualization and manual analysis. So, to fully exploit the rich statistical information available, the recognition should be assisted with an algorithm that operates easily in a multidimensional space.

In this paper, we attempt to develop just such an algorithm. Below, we propose a statistical method based on Linear Discriminant Analysis (LDA) for the automatic recognition of amino acid residue types. It is worth mentioning that this method can be integrated into automatic assignment programs to facilitate the mapping step. Yet, it is not the only application for LDA, as shown in the results section. Of note, other classification methods were also tested (as shown in S1 Text and S1 Fig), but LDA obtained the highest performance scores among all of them. To the best of our knowledge, such a method has only been used once before in protein NMR, to detect beta-hairpin regions [17].

## Materials and methods

### Linear discriminant analysis for the classification of amino acid residue types

The mapping procedure described above can be regarded as a classification problem, as the aim is to assign amino acid residue types to different spin systems. This classification is based on the variables that define spin systems, namely the set of chemical shifts of nuclei belonging to the residue in question, measured in one or more experiments. LDA is a classification method well suited for this purpose.

LDA is related to the more popular method known as principal component analysis (PCA). Both methods look for linear combinations of variables that best explain variance in the data. But while PCA finds new coordinates that maximize the variance of the data, LDA maximizes the variance between the different classes (residue types) and at the same time minimizes the variance within each class [18]. Another important distinction is that in PCA the directions of maximal variance do not depend on the classes, so residue types are not taken into account. By contrast, LDA uses an already classified dataset (the training set) to explicitly attempt to model the difference between the classes. Once adequately trained, the model can then be used to classify the spin systems of an unassigned protein.

An LDA classification model comprises discriminant functions that appear based on the linear combination of predictive variables providing the best separability between classes. These functions are derived from the training set, in which the classifications of the spin systems are known. The training set can be thought of as an $N \times (M + 1)$ table, with NMR data from $N$ spin systems defined by $M$ chemical shift values, where the last column of the table gives the residue types. For the purpose of training the model, the LDA method assumes that the scattering of the chemical shift values of the spin systems belonging to each of the amino acid residue types can be accurately characterized by normal distributions [19]. In this way, the spin systems from the $k$-th residue type are described by a mean vector $\boldsymbol{\mu}_k$ and

a covariance matrix $\Sigma_k$, given by:

$$\boldsymbol{\mu}_k = \frac{1}{n_k} \sum_{i=1}^{n_k} \mathbf{x}_i$$

$$\boldsymbol{\Sigma}_k = \frac{1}{n_k} \sum_{i=1}^{n_k} (\mathbf{x}_i - \boldsymbol{\mu}_k)(\mathbf{x}_i - \boldsymbol{\mu}_k)^\top$$

(1)

where $\mathbf{x}_i$ is a vector representing the $i$-th spin system, with dimensionality equal to the number of chemical shifts (M), and $n_k$ is the total number of spin systems of amino acid residue type $k$ in the training set. To build the LDA classification model, we further assume that the covariance matrices for all classes are the same [18]. This common covariance matrix is usually called a *pooled* covariance matrix, and is given by:

$$\boldsymbol{\Sigma}_{pooled} = \frac{1}{N} \sum_i n_i \cdot \boldsymbol{\Sigma}_i$$

(2)

where the summation runs through all the residue types. If we have a total of 20 amino acid types in the training set, the classification model consists of 20 discriminant functions $\{f_1, f_2, \ldots, f_{20}\}$. These functions define the regions of maximal probability for each residue type in the multidimensional chemical shift space, which may in particular correspond to a single spectrum. Equal probabilities between classes are used as boundary conditions and define hyperplanes that separate each cluster within a class. For new, unassigned spin systems we get probabilities corresponding to each of the amino-acid residue types. The classification score needed for a new spin system $\mathbf{x}$ to belong to residue type $k$ is given by [20]:

$$f_k(\mathbf{x}) = \boldsymbol{\mu}_k^\top \boldsymbol{\Sigma}_{pooled}^{-1} \mathbf{x} - \frac{1}{2} \boldsymbol{\mu}_k^\top \boldsymbol{\Sigma}_{pooled}^{-1} \boldsymbol{\mu}_k + \ln(\pi_k)$$

(3)

where we define the *a priori* probability as $\pi_k = n_k/N$, and $N$ is the total number of spin systems in the training set. Eq 3 is a linear equation on the variable $\mathbf{x}$, and hence the class boundary is a hyperplane of linear shape. Within a single region of this type, a given discriminant function has a higher classification score than all other functions, and all resonances that fall inside this region are classed as belonging to the corresponding residue type.

Importantly, this method assumes that the distribution of the chemical shifts in the mapped protein is similar to that of the training set. For resonance mapping in IDPs, this requirement can be met by choosing, for the training set, proteins described in the BMRB database as "unstructured", "unfolded" or "disordered". Our training set consisted of 1,613 spin systems from 17 such proteins (BMRB Entry IDs: 6436, 11526, 15176, 15179, 15180, 15201, 15225, 15430, 15883, 15884, 16296, 16445, 17290, 17483, 19258, 25118 and 30205). The training set must contain samples from all the amino acid residue types present in the IDP that is to be mapped. Of course, if certain residue types are missing from the protein under investigation, the spin systems corresponding to residues of this type should be removed from the training set, as should all spin systems that lack any of the chemical shifts that we choose to use for discrimination.

Classification models lacking the assumption of Eq 2 are called Quadratic Discriminant Analysis (QDA) models [21]. Initially, one might assume that a QDA model would outperform its LDA sibling. However, in S1 Text and S1 Fig we show that LDA is actually the better choice for protein mapping. In SI we also compare the performance of LDA to two other well-known classification methods: "k-nearest neighbors" and "support vector machines". LDA scores

highest in mean accuracy, sensitivity and specificity (although not by a large margin), while at the same time it grades lowest in variance of accuracy across all 17 training proteins.

To summarize our approach, although other methods can achieve a classification accuracy similar to that of LDA, we choose to use LDA because it demonstrates the lowest variance in classification accuracy after performing cross-validation with different proteins. LDA therefore allows for more consistent predictions across different IDPs.

## Sample preparation

The $^{13}$C,$^{15}$N-uniformly labeled $\alpha$-synuclein was expressed as described by [22]. The sample concentration was 1.35 mM in 20 mM sodium phosphate buffer at pH 6.5. The buffer contained 200 mM NaCl, 0.5 mM EDTA, 0.02% NaN$_3$ and a Protease Inhibitor Cocktail (Roche). 10% D$_2$O was added for lock.

## NMR spectroscopy

All experiments were performed at a temperature of 288.5 K on an Agilent 700 MHz spectrometer equipped with a 5 mm HCN room-temperature probe and DD2 console. The experiments—3D HNCO [23], 4D HNCACO [24], 4D HabCab(CO)NH [25, 26] and 4D (H)N(CA) CONH [26]—were acquired using non-uniform sampling. 3D data was processed using a multidimensional Fourier transform [27] and 4D data was processed using a sparse multidimensional Fourier transform [26], with HNCO as a basis spectrum. Spin systems were formed by gathering data from the peaks appearing on cross-sections corresponding to individual basis spectrum peaks. In the case of overlap of the basis spectrum peaks, when on a given cross-section appeared peaks from more than one spin system, the discrimination of spin systems was performed according to recommendations described in [13]: peaks were regarded as belonging to a cross-section on which they had higher absolute intensity. Using 3D HNCO (instead of $^{15}$N-HSQC) as a basis spectrum made this approach more efficient; amide proton and nitrogen chemical shifts were determined more accurately. Thus, the overlapping cross-sections were more shifted than they would be if 2D basis spectrum was used. The experimental parameters are shown side-by-side in Table 1. All spectra were displayed and analyzed using the Sparky program [28]. The experimental data—raw signals and Sparky spectral files—are available at zenodo.org (10.5281/zenodo.7032142).

## Results and discussion

To use the proposed approach to solve the practical challenges of IDP resonance assignment, we first need to answer the following research questions: Is the training set of BMRB entries mentioned above consistent? Can it be used to classify unknown spin systems? What is the optimal set of chemical shifts that provides efficient discrimination? Can this method assist the chain-mapping step in the assignment procedure? And can it help in other resonance

**Table 1. Experimental parameters: ni (number of hypercomplex increments), nuc (name of nucleus), t (maximum evolution time in ms) and sw (spectral width in Hz).** Experimental time is given in hours.

| Experiment | ni | Dim 1 | | | Dim 2 | | | Dim 3 | | | Exp. time |
|---|---|---|---|---|---|---|---|---|---|---|---|
| | | nuc | t | sw | nuc | t | sw | nuc | t | sw | |
| 3D HNCO | 1600 | CO | 50 | 2800 | N | 75 | 2500 | - | - | - | 12 |
| 4D HNCACO | 2200 | CO | 50 | 2800 | C$\alpha$ | 10 | 6200 | N | 75 | 2500 | 33 |
| 4D HabCabCONH | 1450 | H$\alpha\beta$ | 10 | 7500 | C$\alpha\beta$ | 7.1 | 14000 | N | 75 | 2500 | 22 |
| 4D (H)N(CA)CONH | 2500 | N | 28 | 2500 | CO | 28 | 2800 | N | 75 | 2500 | 40 |

assignment problems, such as peak list transfer between the spectra of samples measured under slightly different conditions? We address these questions in this section.

All the BMRB entries that we used for training contain seven chemical shifts: $H^N$, N, CO, $C_\alpha$, $C_\beta$, $H_\alpha$ and $H_\beta$. Based on these chemical shifts, we constructed three training sets, corresponding to typical experimental setups (see Section Optimal set of chemical shifts): subset (i) $H^N$, N, $C_\alpha$ and CO; subset (ii) $C_\alpha$, $C_\beta$, $H_\alpha$ and $H_\beta$; and subset (iii), which was actually the complete set, containing all seven chemical shifts. Some spin systems in the testing set may be incomplete, that is to say, lacking certain chemical shifts; this is usually the case for the residue preceding the formed chain of spin systems. To classify such spin systems we used a separate, restricted training set consisting of those chemical shifts that are present.

## Consistency of the training data

We evaluated the consistency of the training dataset (17 proteins from the BMRB) by performing a leave-one-out cross-validation using subset (iii): Train the classification model using the NMR data from 16 proteins and test it on the remaining protein, then repeat the process swapping the test protein until all proteins have been used for testing. Fig 3 shows the accuracy of amino acid type recognition by LDA, defined as the number of correct classifications over the total number of classifications. We found the weighted mean accuracy to be 89.43%, weighted by the number of residues in the proteins. It is worth noting that this level of accuracy is high, despite the fact that several of the proteins contained numerous residues lacking one or more chemical shifts values.

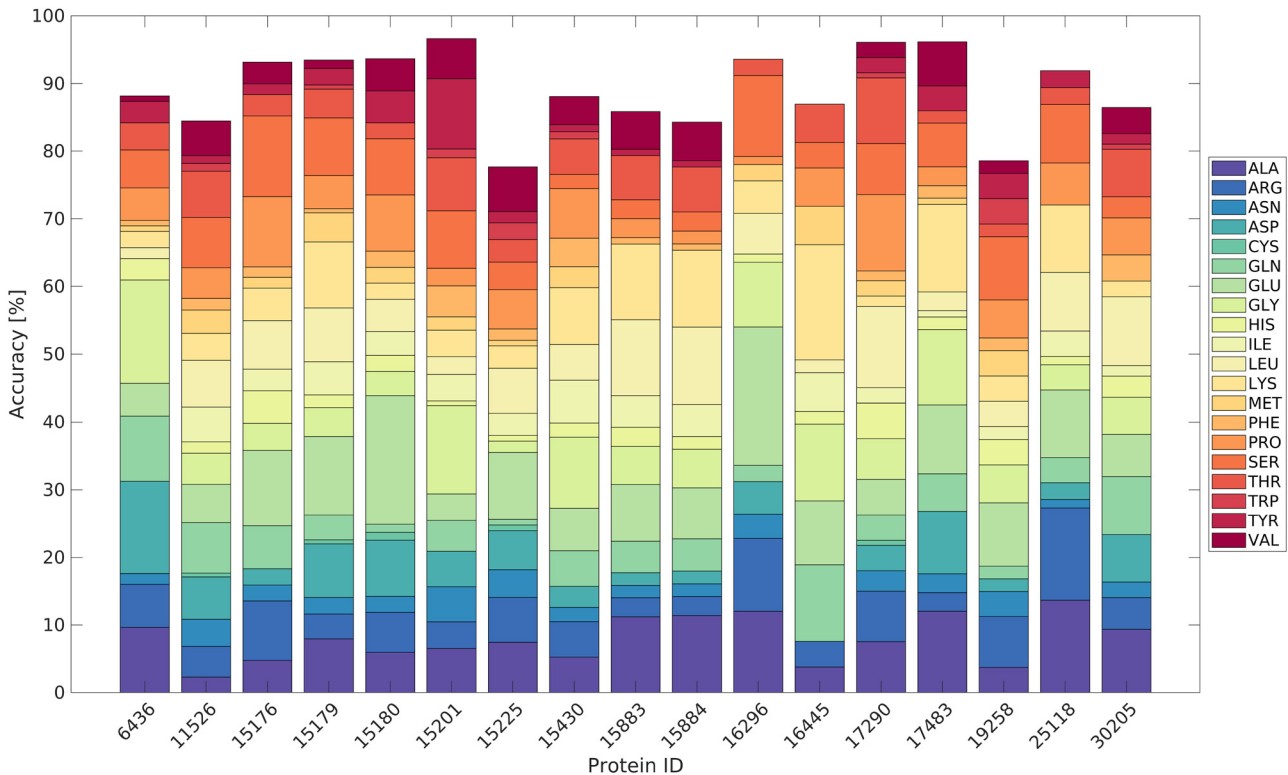

**Fig 3. Classification accuracy for LDA using $H^N$, N, CO, C$\alpha$, C$\beta$, H$\alpha$ and H$\beta$ chemical shifts.** We performed leave-one-out cross-validation with NMR data from the 17 proteins, downloaded from the BMRB. Amino acid distributions are shown as percentages of each type present in each protein.

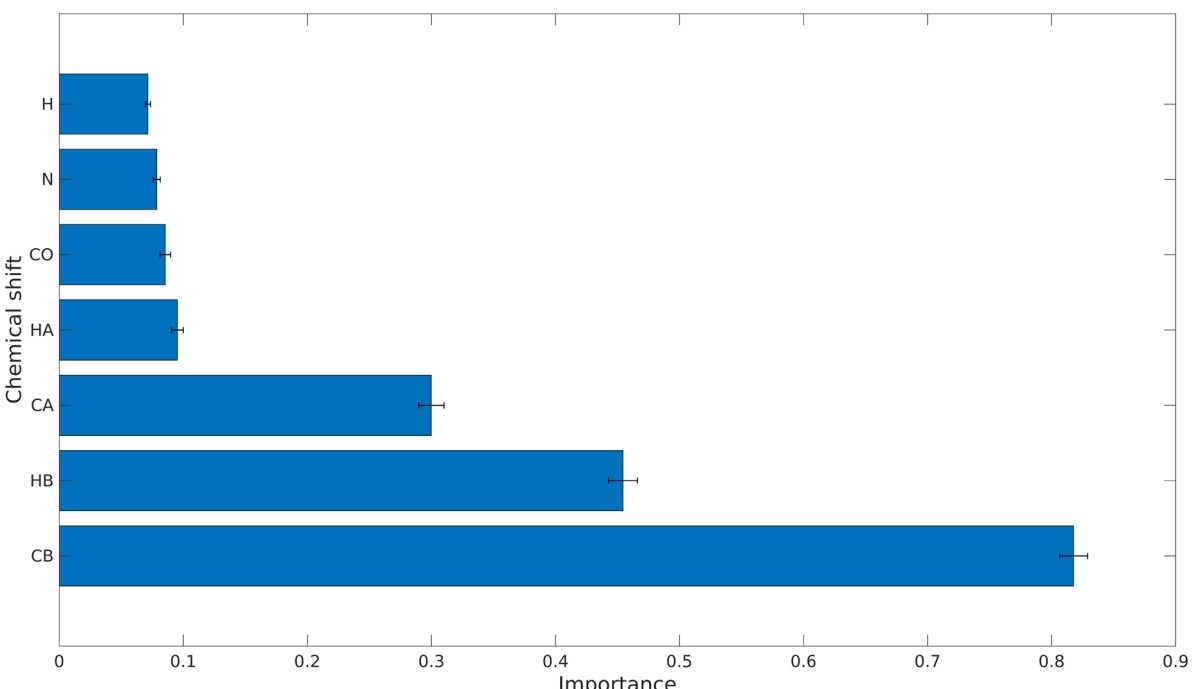

**Fig 4. Importance of different chemical shifts for the classification models, measured as the error rate of LDA models when making classifications on a test set with a shuffled column corresponding to a single chemical shift.** The results shown are the mean values from performing 1,000 tests on each chemical shift, with error bars representing one standard deviation.

This high level of accuracy leads us to the conclusion that the selected BMRB entries do indeed show similar chemical shift distributions for the same residue types. It is therefore not unreasonable to suggest that the chemical shifts of a new, as yet unassigned IDP will be correctly classified using this training data. It is worth mentioning that chemical shift values of training and test proteins are normalized simultaneously using Pareto scaling to correct for possible referencing errors.

## Optimal set of chemical shifts

Some chemical shifts are more important than others in the classification process, that is to say, they have greater predictive power. We measured the importance of the chemical shifts involved to evaluate their impact on the accuracy of the model (Fig 4). To do this, we first trained the model with the complete $N \times (M + 1)$ table, representing the entire training set. Next, we randomly shuffled a single $M$ column (chemical shift value) from the training set. Finally, we used the model that we had previously trained to classify the set with one shuffled column. Randomly shuffling one column removes the correlation between the chemical shift values and the amino acid type for each spin system, while preserving the descriptive statistical information in the column. By measuring the error rate in the classification (defined as 1 minus the accuracy), we can evaluate which chemical shifts are more important for making accurate classifications.

The conclusions in Fig 4 form the basis of the model that we present here. It is well known from statistics [29] that $C_\beta$ and $H_\beta$, followed by $C_\alpha$, are the most characteristic chemical shifts. Yet, the data in Fig 4 shows that the aggregated effect of the other chemical shifts is not negligible, resulting in a complex seven-dimensional pattern that is impossible to analyze manually.

**Table 2. Proposed sets of experiments needed to obtain the described chemical shift subsets (i)-(iii).** In subset (iii), 3D HNCO is required for SMFT processing [26].

| (i) $H^N$, N, C$\alpha$, CO | (ii) C$\alpha$, C$\beta$, H$\alpha$, H$\beta$ | (iii) $H^N$, N, CO, C$\alpha$, C$\beta$, H$\alpha$, H$\beta$ |
|---|---|---|
| 4D HNCACO | 4D HabCab(CO)NH | 3D HNCO |
| 4D HNCOCA | | 4D HabCab(CO)NH |
| | | 4D (H)N(CA)CONH |

When choosing the optimal chemical shift set, we have to take into account the limitations of available NMR experiments. These experiments differ in their dimensionality, sensitivity and established correlations. What we want is a set of experiments providing information about the desired chemical shifts that can be obtained in a minimum amount of experimental time. Our goal is to obtain sets of chemical shifts that characterize different residues. Typically, however, the triple-resonance experiment providing intra-residual correlations also gives additional inter-residual peaks. For example, 4D HNCACO provides the desired correlation $H_i^N$-$N_i$-$C\alpha_i$-$CO_i$ (subset (i)) but it also provides $H_i^N$-$N_i$-$C\alpha_{i-1}$-$CO_{i-1}$. For our purpose, the two types of correlations need to be distinguished by means of an extra experiment, namely 4D HNCOCA, that contains only the latter, inter-residual types of peaks. On the other hand, we can obtain the chemical shifts for subset (ii) with just a single experiment, namely 4D HabCab (CO)NH. In this case, the four chemical shifts (C$\alpha$, C$\beta$, H$\alpha$ and H$\beta$) correspond to the residue preceding the correlated amide group. The experimental setups providing the chemical shifts of the aforementioned subsets (i), (ii) and (iii) are presented side-by-side in Table 2.

We also performed the leave-one-out analysis described in Section Consistency of the training data to evaluate how well the method performed for subsets (i), (ii) and (iii). Fig 5 shows the confusion charts (average LDA probability matrix) for different amino acid types. Even for subset (i), the overwhelming majority of the classifications were correct. However, the results for subset (ii) were much better, that is to say, less ambiguous. The results for subset (iii) are slightly better than for subset (i), but not as good as those for subset (ii). We may therefore conclude that subset (ii) is the optimal choice for LDA analysis.

## Classifying unknown spin systems

We tested the proposed approach on experimental data from $\alpha$-synuclein. Fig 6 shows the peak positions in a C$_\beta$/H$_\beta$ spectral projection, with the chemical shift distribution for the training data set from the 17 BMRB entries superimposed on top of it. Although Fig 4 shows that C$_\beta$ and H$_\beta$ are the most discriminating, they clearly provide insufficient resolution for most of the groups—only alanine, leucine and isoleucine can be clearly distinguished, while the remaining spin systems are impossible to identify based on C$_\beta$ and H$_\beta$ alone. This provides additional motivation for using the more advanced LDA approach.

Fig 7 shows the results of using LDA on the $\alpha$-synuclein data. To make the comparison clear, we used only residues for which all seven chemical shifts were known, allowing us to construct subset (iii). The only exceptions were glycine and proline residues, which naturally lack chemical shifts of missing nuclei (C$_\beta$ and H$_\beta$ for glycine, $H^N$ for proline). These residues were included if all other chemical shifts (5 or 6, respectively) were present. In the figure we did not take into account several residues with resonances missing for any other reason. However, as emphasized previously and shown later in Sections Application 1: Mapping spin-system chains and Application 2: Resonance list transfer, it is generally possible to perform classification of spin systems with missing chemical shifts.

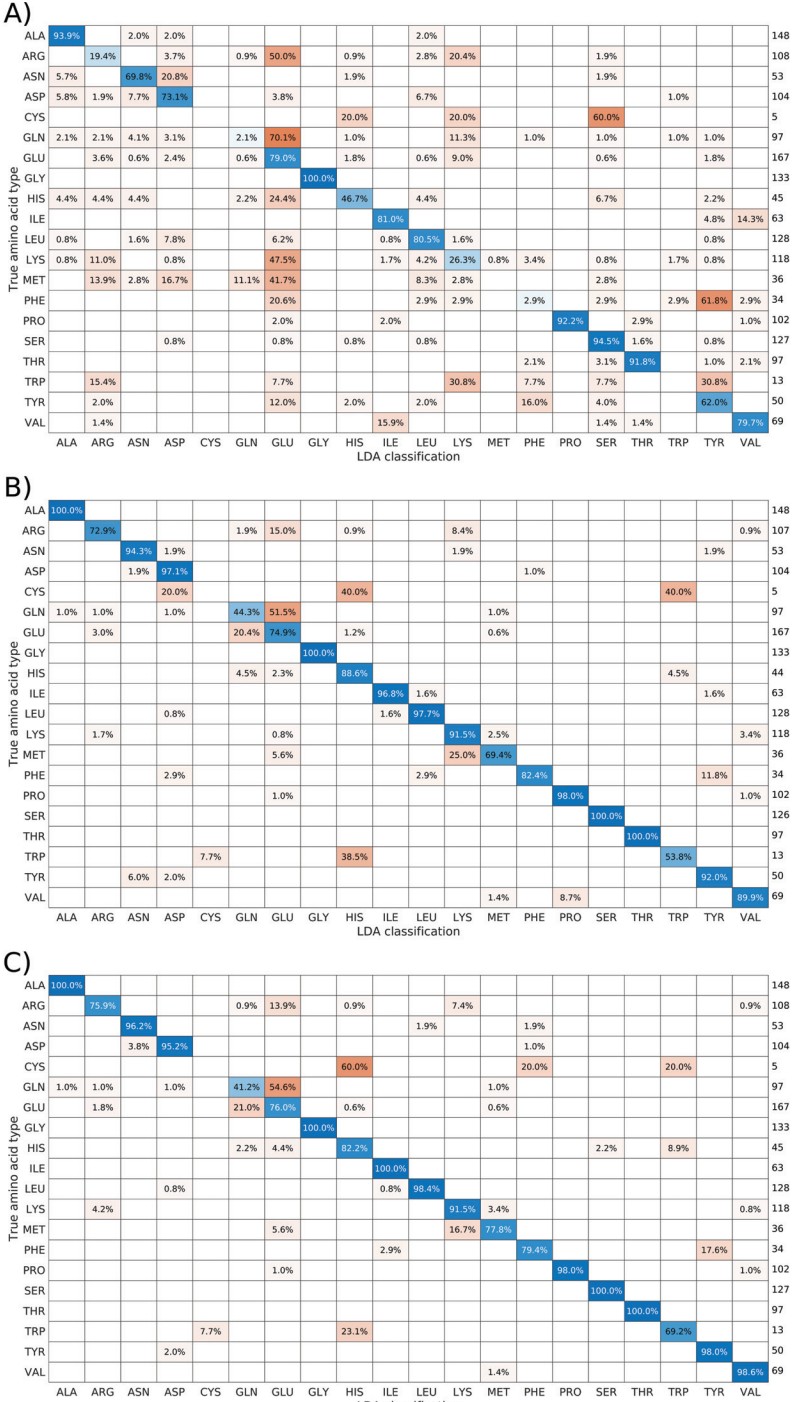

**Fig 5. Confusion charts for the proposed chemical shift sets.** The charts result from performing leave-one-out cross-validation of the 17 proteins from the BMRB used for training. Diagonal elements (in blue) represent correct classifications, and non-diagonal elements (in red) represent incorrect classifications. The values listed to the right of each chart are the sum, for all 17 proteins used, of residues of a given type that were classified. The charts are "row normalized", that is to say, each row shows the distribution of how true amino acid types were classified. Weighted mean accuracy (weighted by the number of residues in the proteins) was calculated for each case: A) subset (i), 66.94%; B) subset (ii), 88.67%; and C) subset (iii), 89.51%.

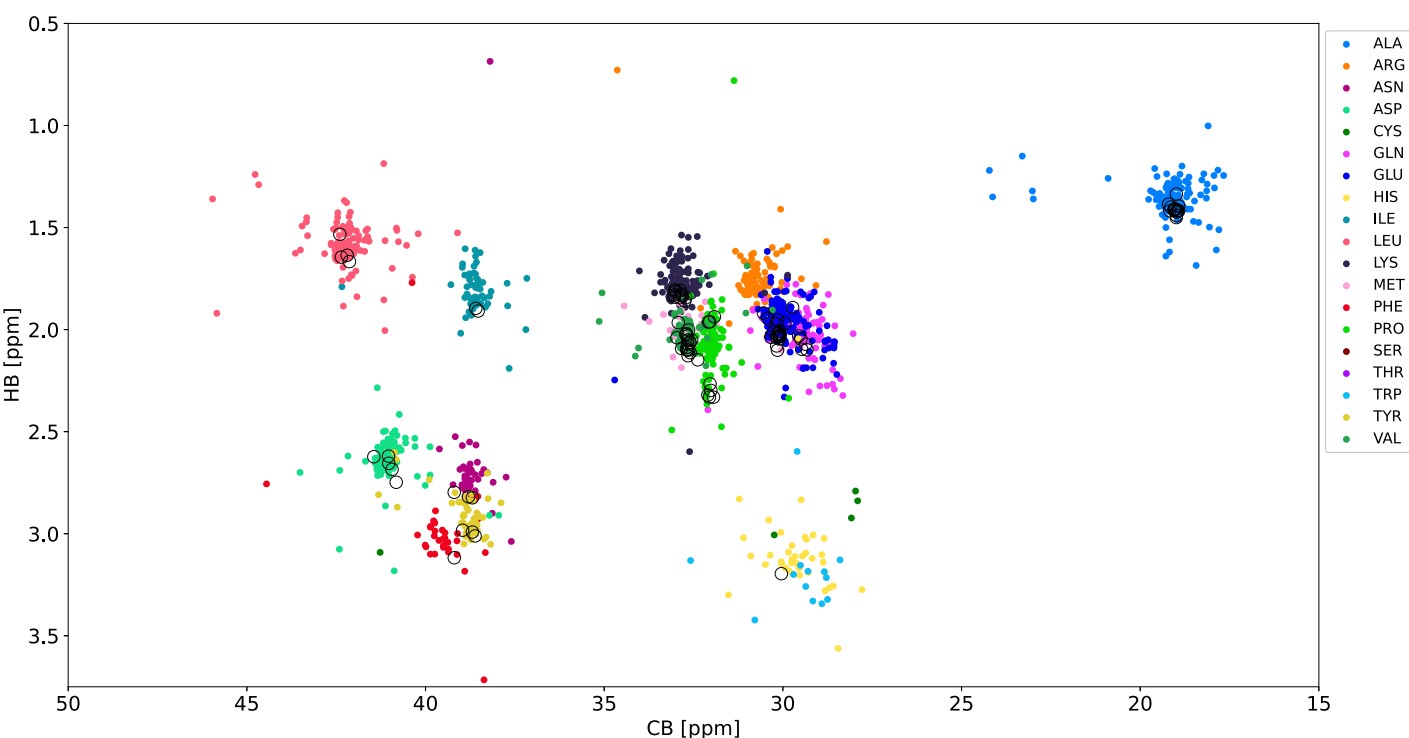

**Fig 6. C$_\beta$/H$_\beta$ plane showing the chemical shifts of 17 unfolded proteins from the BMRB (shown in different colors) and 2D projections of higher-dimensional spectra of $\alpha$-synuclein (shown in black).**

The trends observable in Fig 7 are in line with the predictions given in Fig 4. LDA of subset (i) enables unambiguous recognition of some amino acid types (alanine, serine, threonine and glycine), but it is ambiguous for others. In particular, some groups of amino acid residue types can be confused: a) aspragine and aspartic acid; b) glutamine, glutamic acid and lysine; c) phenylalanine and tyrosine; and d) isoleucine and valine. For these residue types, the probabilities are similar for each of the amino acid types within the group—although, typically, the highest probability corresponds to the correct type. Residues also exist for which the correct amino acid type was not recognized at all using the chemical shifts of data subset (i), namely one methionine, one leucine and one histidine. The ambiguity is reduced when using the chemical shifts of subsets (ii) and (iii), in which case for asparagine, aspartic acid, histidine, isoleucine, leucine and valine, the probability of the identifying the correct amino acid type is almost 100%. For lysine, methionine and tyrosine, the probability exceeds 70%. The only ambiguities that remain are glutamine and glutamic acid (although now the probability of identifying the correct type are higher) and phenylalanine (that can still be confused with tyrosine).

We may conclude that subset (ii) provides results that are almost as good as those for subset (iii), and that both subsets allow much more effective LDA than subset (i). As subset (ii) can be obtained from a single experiment, we recommend using this subset as the approach of choice. The python code used for LDA analysis along with NMR data of the $\alpha$-synuclein IDP and all input files needed to reproduce the results shown in Fig 7 are readily available at a public GitHub repository [30].

## Application 1: Mapping spin-system chains

The assignment of backbone resonances in a protein is a two-step process: First, we form spin system chains, then we map them onto the known amino acid sequence. The latter step can be

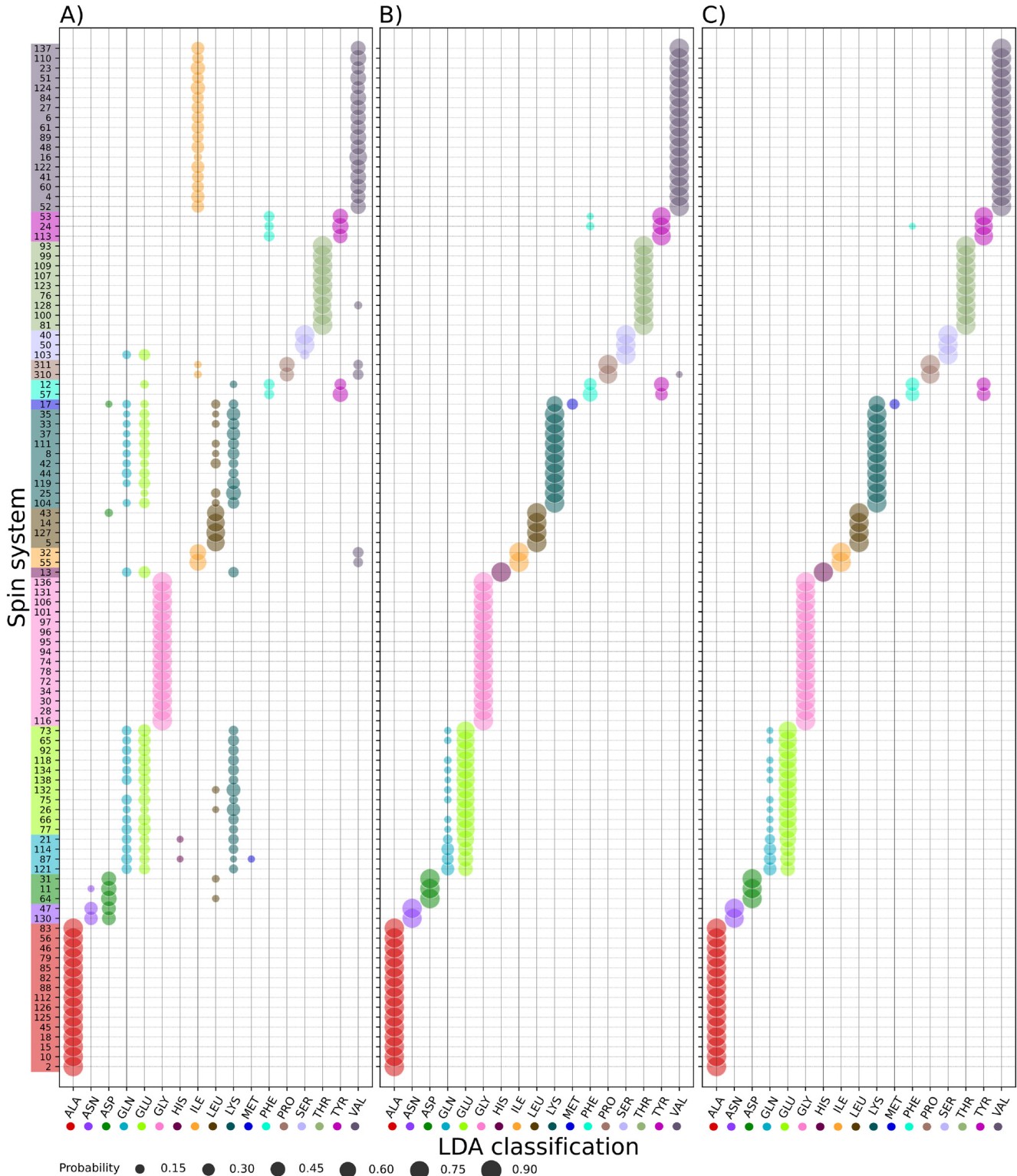

**Fig 7. Results of LDA of the chemical shifts of $\alpha$-synuclein: A) subset (i) ($H^N$, N, CO and $C_\alpha$); B) subset (ii) ($C_\alpha$, $C_\beta H_\alpha$ and $H_\beta$); and C) subset (iii) ($H^N$, N, CO, $C_\alpha$, $C_\beta$, $H_\alpha$ and $H_\beta$).** The vertical axis shows the spin system numbers, the horizontal axis the amino acid types (LDA classes). Spin system numbers are colored according to their true amino acid type. Marker sizes indicate the probability, according to LDA, that the spin system in question belongs to a given class.

greatly enhanced by LDA. LDA identifies residues in a chain more efficiently than traditional "manual" recognition, which typically finds only glycines, alanines and serines/threonines. Optionally, the LDA analysis can be followed by filtering the results using an amino-acid sequence of the protein under consideration. Filtering "impossible" chains can be done automatically by using the output of the LDA analysis. First, for each chain a number of amino acid sequences are formed, which rise from all the combinations of amino acid types that LDA predicts as probable for each spin system in the chain. Then, combinations which are not present in the sequence of the test protein are discarded. This procedure is included in the code provided in the GitHub repository as an optional feature: if an input file containing the spin systems chains is given as input, then the code will give an extra spreadsheet as output detailing all possible amino acid sequences for each chain, their probabilities and the discarded combinations. In the examples below we did use this option.

Fig 8 shows chain mapping cases of increasing difficulty. The easiest task is to map relatively long chains with several easily recognizable residues. The chain of seven residues shown in Fig 8A) contains two glycines and can be mapped manually without ambiguity. Nevertheless, LDA provides even more reliable mapping, as it recognizes all seven residue types with > 90% probability.

Fig 8B) shows the more difficult case of a shorter chain. It is possible to manually identify one of the three residues as serine or threonine. However, this is not sufficient for unambiguous mapping. On the other hand, LDA followed by amino acid sequence filtering provides precise result.

Sometimes, as in Fig 8C), the probability corresponding to the correct amino acid type is not the highest one. Fortunately, we can make the correct choice based on a protein sequence that lets us rule out "impossible" chains, even if the LDA implies that they are the most probable ones. Notably, unambiguous manual mapping of the chain in Fig 8C) is not possible, as only one characteristic residue (glycine) is present.

Often, short chains do not contain even a single easily recognizable residue. Fig 8D) gives an example of such a chain, one that is practically impossible to map manually. LDA makes mapping possible, but we need to consider various combinations of residues. In the case in Fig 8D), only one combination—that with the second-highest LDA probability for two of the three residues—corresponds to the fragment of the protein sequence allowing unambiguous mapping.

Finally, in rare cases, LDA may produce the wrong result for specific residues. An example is shown in Fig 8E). The "preceding" residue (6K) is wrongly identified by LDA as glutamic acid or glutamine; in fact, it is lysine. It could not be glutamic acid or glutamine, as this would lead to "impossible" chains (EGLS or QGLS) that do not exist in the protein sequence. In such cases correct mapping would require manual intervention or employment of more advanced assignment algorithm. Notably, in this example it is again impossible to unambiguously map the chain manually, as with one glycine, three mappings are possible.

## Application 2: Resonance list transfer

Another example of where LDA can be used is the common task of transferring a resonance list from the repository (for example, the BMRB) to the experimental spectrum of a new sample. Usually, the experimental conditions (temperature, pH, ionic strength, concentration, and so on) are not exactly the same as those reported in the repository, and peaks may be shifted in a non-systematic manner. To illustrate this problem, we investigated one of the regions of the $^{15}$N HSQC spectrum of $\alpha$-synuclein, containing a variety of amino acid residue types (see Fig 9A).

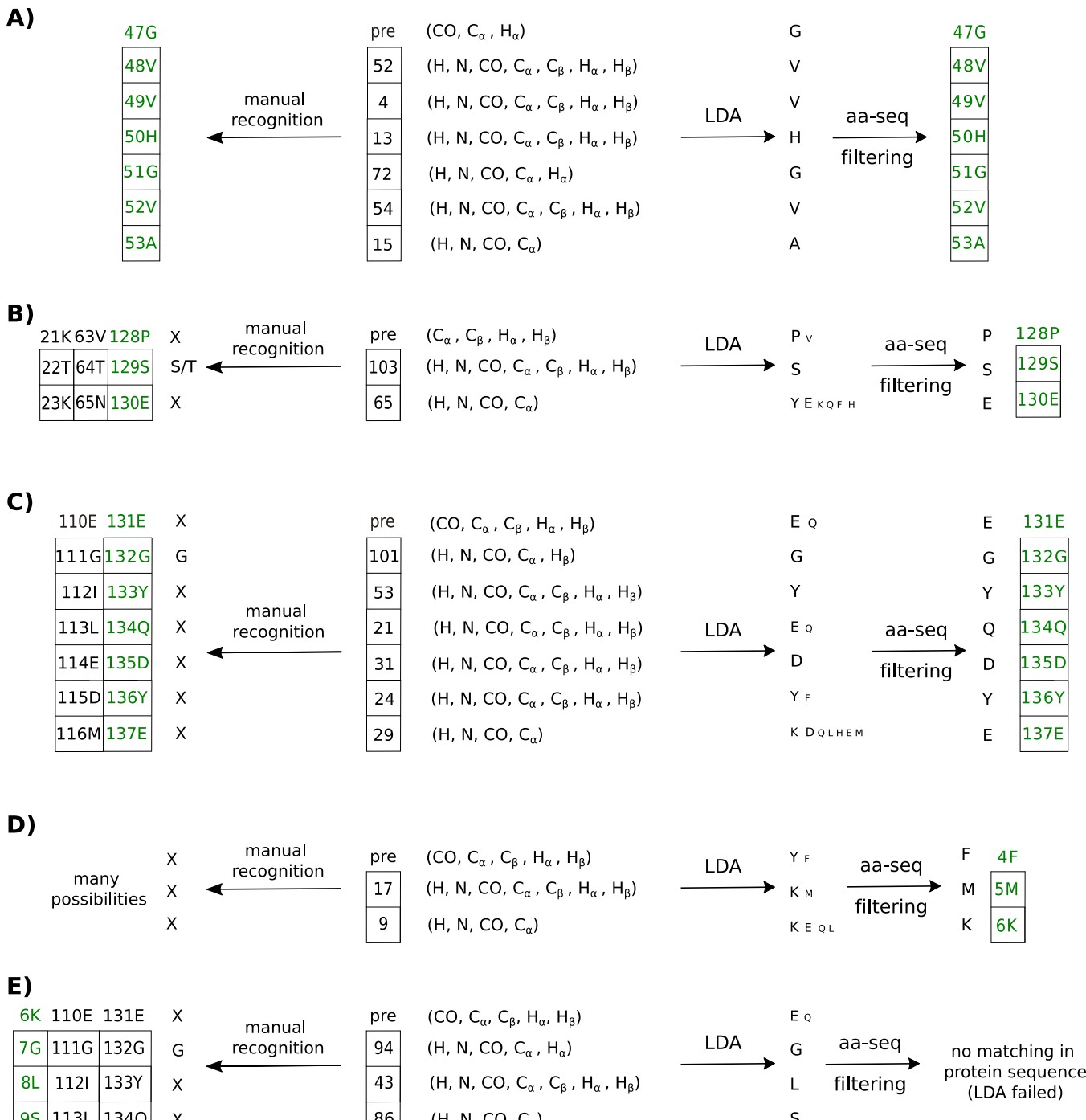

**Fig 8. Comparison of the chain mapping procedure for *α*-synuclein with and without LDA analysis.** LDA was performed using subset (iii), but in some residues, certain chemical shifts were missing and a reduced dataset was used. The numbers of the spin systems forming the chains are shown in square boxes. The label "pre" stands for the amino acid residue preceding the formed chain. The arrows labeled "LDA" point to the results of the LDA analysis, and the size of the one-letter codes corresponds to the probability as determined by LDA. The arrows labeled "aa-seq filtering" point at the results after amino-acid sequence filtering, reducing the number of possibilities. The arrows pointing left point to unambiguous manual identification of glycine, alanine, serine and threonine ("X" is used for all other amino acid types). All chains for which identification is consistent are shown on both sides, and the correct chain is marked in green. Panels A)-E) correspond to chain-mapping tasks of increasing difficulty (see Section Application 1: Mapping spin-system chains for details).

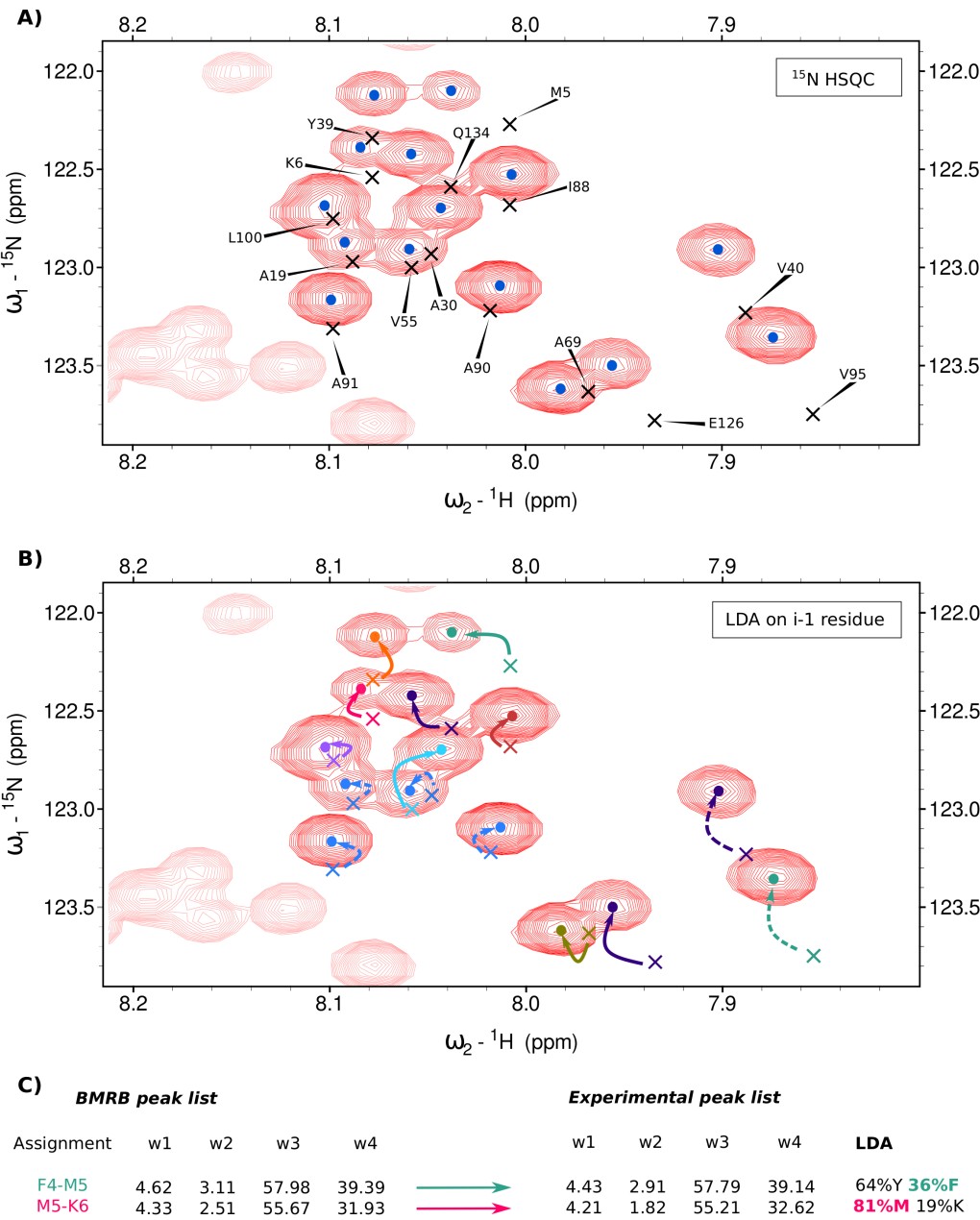

**Fig 9. Application of LDA to the resonance list transfer from BMRB Entry 6968 to the experimental spectrum of α-synuclein.** A) A fragment of the $^{15}$N-HSQC spectrum with peaks marked with dots. Peaks from the BMRB list are marked with crosses. Labels indicate the corresponding residue name. B) Correct transfer of assignment from BMRB to the experimental peak list (unambiguous assignments shown with solid arrows, more ambiguous ones—with dashed arrows). The experimental and BMRB peaks corresponding to the successors of the residues of the same amino acid type are

marked in the same color. C) $H_\alpha$, $H_\beta$, $C_\alpha$ and $C_\beta$ chemical shifts of the residues preceding the residues shown in panels A) and B). For experimental peaks, these chemical shifts were obtained from the 4D HabCab(CO)NH experiment and then analyzed with LDA, enabling assignment transfer. Color coding as on panel B). The full spectrum with both peak-lists is available at 10.5281/zenodo.7032142.

We picked the 16 resonances in the region of interest (indicated by dots in Fig 9A). We also loaded the peak list from the BMRB entry 6968 (indicated by crosses). It appears that almost all peaks from the BMRB list deviated from the peaks generated in the experiment, resulting in ambiguity during transfer of assignment.

For this reason, we decided to facilitate the transfer by means of LDA. Using $^1$H and $^{15}$N experimental peak positions, we peak picked the 4D HabCab(CO)NH spectrum, obtaining the $H_\alpha$, $H_\beta$, $C_\alpha$ and $C_\beta$ chemical shifts of the preceding residues. We then performed LDA on them (Fig 9C).

In many cases, where only one resonance corresponding to the given amino acid type occurred in the proximity of the peak under consideration, LDA recognition allowed for unambiguous transfer of assignment (solid arrows in Fig 9B and 9C). However, where several peaks corresponding to the same type occurred close to the peak, ambiguity remained (assignment shown with dashed arrows). For instance, in the region considered in Fig 9A there were four alanine peaks. The patterns of the experimental and BMRB peaks corresponding to this amino acid type (indicated by blue dots and crosses in Fig 9B) were very similar, and indeed, in this case, choosing the nearest option was correct. However, care is called for in such situations, as some deviations may be significant; in the absence of additional information (for example, about the sequential connectivities) the mapping may be incorrect.

An interesting case is the BMRB peak corresponding to Y39-V40. Two experimental peaks occur in its vicinity, both with significant probabilities of being a tyrosine residue at $i-1$ position (43% in the case of the closest experimental peak, 89% in the case of the second-closest peak). However, another BMRB peak also occurs close by, corresponding to F94-V95. As the probabilities of phenylalanine for the experimental peaks in question were 56% and 11%, the F94-V95 was assigned to its closest peak and Y39-V40 to its second-closest peak. Thus, the probabilities represent valuable information in such cases.

To sum up, using LDA can greatly facilitate assignment transfer, often preventing incorrect transfer to the closest neighboring peak. However, even when using LDA, care is called for and different possibilities should be considered.

## Practical recommendations

Our experiments show that LDA is a versatile tool that can support spectral analysis in many ways. Users will no doubt develop their own habits and practices when it comes to employing the tool. Below, we summarize our own practical recommendations, based on our experience.

- As in all machine learning methods (MLMs), the best training set for LDA will have similar features to those of the test data. For the chemical shifts of proteins, that means that all the molecules in the training set and the test data should lack a secondary structure. This condition is not strict, however, and—as shown in this paper—quite impressive results are possible even with a very coarse selection of proteins for the training data.

  In fact, the training data contained small regions with residual structure: alpha-helical (up to 50%) for residues 60–78 in BMRB data set 11526, similarly long (19 residues) $\alpha$-helical linker in set 15176 and several transient helices in data set 15179 (one of them with secondary shifts of $C_\alpha$ of up to 4 ppm, suggesting complete formation of the helix).

- As many training spin systems as possible should be used (again, this is true for all MLMs). Interestingly, for very large proteins it might be possible to use data from the same molecule for both training and testing. In other words, LDA can be used to complete the assignment after the majority (for example, 80%) of the residues have been assigned using traditional methods, and then used to form the training set.

- 4D HabCab(CO)NH is the experiment that provides the best data for LDA. For optimal resolution, non-uniform sampling (NUS) must be used for signal acquisition, with a variety of possibilities for processing. As we had a 3D HNCO at our disposal, we were able to process the data using a sparse multidimensional Fourier transform [26, 31], but numerous other options are possible, including compressed sensing [32, 33], maximum entropy [34], variants of the CLEAN algorithm [35, 36], projection spectroscopy [37], and many others [38–41]. The proper separation of NH resonances is particularly important, since it affects the proper determination of the most differentiating chemical shifts ($C_\beta$, $H_\beta$) and formation of spin systems. Besides resolution-enhancement by NUS, one may solve the problem by increasing the experiment dimensionality—acquisition of 5D HabCabCONH would allow separating the spin systems using triples of frequencies ($H_i^N$, $N_i$ and $CO_{i-1}$) which significantly reduces the overlap problem. [25, 26] Another approach can be the application of methods based on $^{13}C$ detection. [42] On the other hand, the ambiguous cases that result from peak overlap can be easily detected and not taken into consideration. Thus, the situation is not as "dangerous" as e.g. for the sequential assignment.

- The results of the LDA should always be combined with other available information. For example, incorrect classifications can often be detected by comparing the results from LDA for the spin system chains with the protein's primary structure. "Impossible" chains should not be considered, even if the amino acids that compose it have the highest probability according to LDA. Automatic filtering of impossible chains is incorporated as an option in our code.

## Conclusion

In this paper, we show that linear discriminant analysis (LDA) is a reliable classification method supporting the assignment of resonances in NMR spectra. The method can help with many tasks, such as chain mapping and assignment transfer. In our experience, it is easy to obtain a training set from the BMRB with only coarse filtering—that is to say, using proteins marked as "unfolded", "unstructured" or "disordered". We believe that, with a growing number of IDP resonance assignments in the databases, this method will become even more powerful and reliable in the future.

## Supporting information

**S1 Text. LDA vs. other classification methods.** We analyzed the performance of LDA, Quadratic Discriminant Analysis (QDA), K-Nearest Neighbours (KNN) and Support Vector Machines (SVM) to pick the best classification method for our research. We looked at specific performance parameters such as accuracy, sensitivity, specificity and consistency.
(PDF)

**S2 Text. LDA classification of proteins in the training set.** The results of LDA for the proteins from the training data obtained in the same way as Fig 7 (subset (iii)).
(PDF)

**S3 Text. Comparison of LDA performance with TSAR amino-acid recognition procedure.**
The performance of LDA approach was compared with the amino-acid recognition procedure of the TSAR program. [13]
(PDF)

**S1 Fig. LDA classification of proteins in the training set.**
(PDF)

**S2 Fig. Sensitivity and specificity of the methods.** Comparison of performance in classification for all 4 methods.
(PDF)

**S3 Fig. LDA classification of proteins in the training set.** The results for all 17 proteins from the BMRB that compose the training set are shown, demonstrating the efficiency and accuracy of the LDA approach.
(PDF)

**S4 Fig. Comparison of LDA performance with TSAR amino-acid recognition procedure.**
The results were shown for $\alpha$-synuclein spin sytems.
(PDF)

**S1 Table. Summary of classification performances.** Values of analyzed classification performance parameters are given for each method.
(PDF)

**S2 Table. Training set sample conditions.** Sample conditions of the training data sets, as reported in BMRB entries.
(PDF)

## Acknowledgments

The authors thank Dr. Thomas Schwarz and Prof. Robert Konrat from the University of Vienna, Max Perutz Laboratories for providing the $\alpha$-synuclein sample.

## Author Contributions

**Conceptualization:** Krzysztof Kazimierczuk, Anna Zawadzka-Kazimierczuk.

**Data curation:** Javier A. Romero, Paulina Putko, Krzysztof Kazimierczuk, Anna Zawadzka-Kazimierczuk.

**Formal analysis:** Paulina Putko, Krzysztof Kazimierczuk, Anna Zawadzka-Kazimierczuk.

**Funding acquisition:** Krzysztof Kazimierczuk.

**Investigation:** Javier A. Romero, Paulina Putko, Anna Zawadzka-Kazimierczuk.

**Methodology:** Javier A. Romero, Krzysztof Kazimierczuk, Anna Zawadzka-Kazimierczuk.

**Project administration:** Krzysztof Kazimierczuk.

**Software:** Javier A. Romero, Mateusz Urbańczyk.

**Supervision:** Krzysztof Kazimierczuk, Anna Zawadzka-Kazimierczuk.

**Validation:** Javier A. Romero, Anna Zawadzka-Kazimierczuk.

**Visualization:** Javier A. Romero, Paulina Putko, Mateusz Urbańczyk, Anna Zawadzka-Kazimierczuk.

**Writing – original draft:** Javier A. Romero, Paulina Putko, Krzysztof Kazimierczuk,
Anna Zawadzka-Kazimierczuk.

**Writing – review & editing:** Paulina Putko, Krzysztof Kazimierczuk,
Anna Zawadzka-Kazimierczuk.

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
