## [Decision Letter · Decision Letter 0]

14 Jul 2022

Dear Prof. Kazimierczuk,

Thank you very much for submitting your manuscript "Linear discriminant analysis reveals hidden patterns in NMR chemical shifts of intrinsically disordered proteins" for consideration at PLOS Computational Biology.

As with all papers reviewed by the journal, your manuscript was reviewed by members of the editorial board and by three independent reviewers. In light of the reviews (below this email), we would like to invite the resubmission of a significantly-revised version that takes into account the reviewers' comments. **Importantly, please make your software, user manual, training and test sets available and include a GitHub link in the abstract. **

We cannot make any decision about publication until we have seen the revised manuscript and your response to the reviewers' comments. Your revised manuscript is also likely to be sent to reviewers for further evaluation.

Sincerely,

Anna R Panchenko

Associate Editor

PLOS Computational Biology

Arne Elofsson

Deputy Editor

PLOS Computational Biology

Reviewer's Responses to Questions

**Comments to the Authors:**

Reviewer #1: The paper presents an interesting method for determining residue types in the NMR backbone assignment of intrinsically disordered proteins (IDPs), which generally suffer from severe resonance overlap. Linear discriminant analysis (LDA) is used to recognise amino acid residue types and shown to work (slightly) better than three other approaches.

The method is useful, and the paper is very well written and illustrated. The caveat is that the method works on ‘spin systems’ that have to be assembled beforehand from spectra/peak lists. This can be a limiting factor for IDPs, which generally feature extensive overlap that precludes the unambiguous formation of spin systems.

Provided that the software is made available (which seems not to be the case now), I propose publication after (very) minor revision.

Minor points:

- Author summary, p.1: NMR is one of the methods for IDP research, rather than “the method”.

- Methods (or Results): Describe the preparation of spin systems, in particular, how ambiguities have been handled/resolved.

- p. 5: The results of the comparison of four classification methods in the SI is hardly discussed in the main paper.

- p. 6: ‘(see Section )’ misses the section number.

- p. 7: ‘described in Section ‘ misses the section number.

- p. 8: phenyloalanine —> phenylalanine (2x)

- p. 9, Application 1: Again, describe how spin systems are formed.

- p. 9, caption of Fig. 8: From which protein(s) are the examples shown?

- P. 10: ’16 resonances in the region of interest’: Is the situation similar the other residues?

- Ref. 35 Wu K —> Wüthrich K

- Ref. 36 Orekhov VYVY —> Orekhov VY

- Fig. 6: What are the outliers at w(HB) = 0.6-0.8 ppm?

Reviewer #2: The Romero et al present a statistical approach to facilitate backbone assignments of IDPs. Whilst the work contains some interesting ideas, there are a number of issues and limitations.

1. The authors somehow failed to demonstrate that the approach have any advantages over the use of the 'traditional' chemical shift connectivity methods along that are very successful to assign small and well-behaving IDPs as 1 mM aSyn. The importance of the work would be significantly enhanced if the author could demonstrate that the proposed algorithm improves accuracy and/or time-efficiency for more complex IDPs; for examples, IDPs with limited/sparse chemical shift connectivities (e.g., due to low sample concentrations or poor data quality) or/and with significant peak overlapping (e.g. 300+ a.a. IDPs).

2. The figure 5 clearly demonstrated that the accuracy of amino type predictions is <70% for 7 a.a. types out of 20, even if all seven chemical shifts were used for the analysis. The authors should provide specific examples to demonstrate how this (sometimes inaccurate) information could be incorporated into the assignment pipeline and how the use of this information would benefit the accuracy and/or efficiency the assignment process.

3. The proposed methods obviously relies on the availability (particularly CB) chemical shifts for individual spin systems. However, for the larger (100+ a.a.) IDPs the unambiguous assignments of 3D and 4D peaks to a specific spin systems often became a major issue due to peak overlapping in NH 2D. The authors should discuss how this would affect the accuracy of the prediction.

4. In Application 2, the authors should show the accuracy of assignment transfer for all peaks in the spectra rather than for a small regions as well as compare it with the accuracy of assignment transfer obtained using only connectivity information from the 4D HabCac(CO)NH spectrum. Moreover, I expect that for aSyn, the assignments for the majority of resonances can be unambiguously transfered using connectivities obtained from a simpler (and more sensitive) 3D HNCA experiment, while CB chemical shift would be critical for the LDA approach. The authors should provide example(s) that clearly demonstrates that the information about a.a. types (in addition to connectivity information obtained from the same experiment) would result in more accurate/efficient assignment transfer.

5. The authors should provide detailed information about the training protein set they used, including protein sizes, a.a. composition, experimental conditions (particularly, pH, temperature, referencing method).

Reviewer #3: --Romero et al. have outlined a new approach to assign spin systems to the amino-acid sequence of a protein, which is one of the most important steps in every bioNMR project. The approach, based on linear discriminant analysis (LDA), enables both resonance assignment as well as the transfer of assignments to spectra obtained under slightly different conditions where the peaks have moved from the reference assignment. I am impressed with the unique approach used by the authors, as I was previously unaware of LDA and had not seen it applied in NMR. That said, I am not surprised that these authors developed this new approach, given their excellent track record of applying NMR to obtain quantitative insight into intrinsically disordered proteins as well as developing advanced non-uniform sampling/spectral reconstruction methods and automated resonance assignment methods (e.g., TSAR).

--The new LDA approach performs very well on the tested benchmark comprising 18 proteins from the BMRB with known resonance assignments, with a mean weighted accuracy of 90%. I note that the accuracy per protein ranges from ~98% to ~78%, so there is room to explore why some proteins yield poorer results (see below). Moreover, the new LDA approach not only provides highly accurate resonance assignments based on a single 4D HabCab(CO)NH spectrum (recorded with NUS), but also shows the most probable assignment followed by the next most probable assignments. This is very useful for the user, as for spin systems with confidence values that are not close to 100%, the user can consider the top 2-3 most probable assignments to manually intervene.

--In general, I am strongly in favor of publishing this work in PLOS Computational Biology, given that the manuscript outlines an innovative approach, contains new methodology, and has the potential to advance the bioNMR studies of all IDPs (as well as folded proteins, given a re-training on folded protein spectra). Thus, I congratulate the authors on their new algorithm and the unique insight that it brings. However, before publication, I would suggest to the authors a few changes that would hopefully strengthen their work and demonstrate the rigor and utility of this new LDA approach. All of my suggested changes should be relatively easy to do, and so I would welcome a resubmission after the authors have addressed these minor concerns.

--I have outlined my suggestions below:

(1) Given that this new LDA approach seems to have most utility in the resonance assignment procedure, the authors should compare the performance of their LDA approach with some other automatic resonance assignment software (e.g., MARS, TSAR, etc.). An ideal test scenario would be a performance comparison on alpha-synuclein given the same input chemical shifts (e.g., from the single 4D HabCab(CO)NH). Also, it would also be interesting to compare their performance of e.g. MARS or TSAR or AUTO-ASSIGN on the BMRB benchmark of 18 proteins with the results from the LDA approach.

--The main strength of the LDA approach, as I see it, is the LDA classification percentages (e.g., Figure 5 and Figure 7), which provides the user with more than one option in cases of low confidence. Thus, other software that only return one solution may yield inaccurate solutions whereas the LDA approach would very easily identify some ambiguity in the assignment solution space, which could most likely be easily resolved (as shown in Figure 8). Thus, it is important to show that the LDA approach provides new information relative to other state-of-the-art automatic assignment approaches.

(2) In Figure 8 (p. 9 of the text), the authors discuss how the LDA approach provides probability distributions in which one class (AA type) may not dominate (i.e., there is assignment ambiguity). The authors then state that the protein sequence can be used to rule out impossible solutions that cannot be consistent with the proposed assignments and the primary structure.

--I wonder if the authors could incorporate this manual step into the automatic analysis procedure? For example, if this manual calculation could be included as an automatic post-analysis step that follows the initial LDA classification, it would be helpful to the user. Because I fear that uninformed users will simply take the highest assignment probability as the “answer” and not check for compatibility with the primary structure/chemical shifts.

--If this analysis of probability distributions could be coded as consistency check with the primary structure/chemical shifts, it would be a nice extension of the work to ensure that the assignment ambiguity does not lead to errors but rather is exploited by the program to circumvent errors. Of course, there will be situations where the probability distribution analysis followed by a post-processing step of consistency with primary structure (which the authors currently perform manually on a case-by-case basis) cannot return an unambiguous correct result. In these cases, the LDA approach shines because it has informed the user that multiple solutions fit the data equally well and there may be the need to collect additional NMR data or to manually inspect the data very closely. Having the two steps combined into one automatic approach (i.e., LDA analysis + inspection of consistency with primary structure) would further strengthen the confidence in the results, knowing that the consistency check may be able to prevent erroneous interpretation of the results by the users.

(3) The authors assume one normal distribution for the chemical shift of a given spin in a given residue. And the authors further assume that the input chemical shifts in the benchmark would be representative of the underlying chemical shift distribution in the protein under investigation. However, the authors’ assumption of a single normal distribution may be problematic in the scenario in which the IDP under investigation has significant residual structure, meaning the distribution would have to cover +/-4 ppm. Residual secondary structure occurs frequently in IDPs, and a quick browsing of the BMRB benchmark used by the authors shows that BMRB ID 11526 has nearly significant alpha-helical secondary structure between residues ~60-78, reaching a maximum value of ~50% near residue 70 (https://link.springer.com/article/10.1007/s12104-013-9523-1). Another transient helix is found with secondary 13CA shifts of ~2 ppm in Darrp-32 (https://pubs.acs.org/doi/10.1021/bi801308y) from BMRB ID 15176 in the benchmark. Finally, in BMRB 15179 of the benchmark, there are at least 3 transient helices, the most C-terminal of which has secondary 13CA values near 4 ppm (from the same paper) suggesting near complete formation of the helix.

3a. Given these large secondary CA values (and presumably correspondingly large HA, HB, CB, CO secondary shifts going in the appropriate directions), could the authors comment on how they expect deviations from random coil character to affect the performance of the LDA classification?

3b. I wonder if there is a correlation between the errors produced by LDA analysis and the secondary CA or CB shifts? That is to say, have the authors noticed that the LDA classifier fails in situations where there is significant residual structure? This might be expected if the majority of residues in the benchmark are random coil whereas a small minority adopt transient secondary structure.

3c. Were the input chemical shifts corrected for possible referencing errors? I wonder if using, e.g. a sequence-based prediction of the expected/neighbor-corrected random coil shift (e.g., POTENCI) might yield a more accurate “center” of the normal distribution for the expected shift given the solution pH and temperature.

(4) On p. 8 the authors note that residues such as Glycine and Proline (and others with missing chemical shifts) were not used in the LDA classification procedure for alpha-synuclein. However, I worry that the authors might be discarding otherwise potentially useful data – for example, Gly residues are often used manually to make the first assignments given that XG/GX motifs in the sequence may be readily identified. And Pro residues have relatively unique chemical shifts that render them readily identified in the LDA classification shown in Figure 5B and 5C. In these plots, Gly is identified with 100% accuracy and Pro with 97% accuracy – thus, these two residues are highly useful and can be identified by LDA with high accuracy.

4a. Could the authors somehow include Gly and Pro in the calculation rather than excluding them?

4b. Along the same lines as above, could the authors make use of non-Gly/Pro residues that have incomplete data? For example, in Figure 4 I note that the authors show that CB is by far the most important for classification followed by HB and CA. The remaining chemical shifts are less important (< 0.1 relative importance). Thus, for residues that have at least CB HB and CA shifts, I would assume that these can still be used as input?

5. Finally, given that the most important shifts for the LDA classification are CB and HB, and thus side-chain shifts, I wonder if the authors considered using C(CO)NH or H(CCO)NH spectra? Was there a particular reason to choose the 4D HabCab(CO)NH spectrum over those mentioned above? I ask because the C(CO)NH and H(CCO)NH spectra would not only provide the highly diagnostic CB and HB shifts, but also even more diagnostic shifts for residues that extend beyond CB/HB. https://www.sciencedirect.com/science/article/abs/pii/S1064186683710198?via%3Dihub

For example, I note that Fig 5 shows that the LDA classification yields confidence values below 80% for Arg, Cys, Gln, Glu, Met, Phe, Trp, and Tyr residues. All of these except Cys would be very readily resolved with access to spins beyond HB/CB. I note that the authors have published an elegant 5D HC(CC-TOCSY)CONH that would provide access to some of these shifts.

https://www.sciencedirect.com/science/article/pii/S109078070900007X?via%3Dihub

If the problem lies with poor access to chemical shift statistics of IDPs for spins beyond HB/CB, then that is indeed a fundamental problem. But given that the LDA classifer performed so well with only 18 proteins as training input, I would hope that there are at least 18 IDPs that have complete 1H and 13C side-chain assignments to enable LDA classification with this expanded benchmark for side-chain chemical shifts.

Minor comments

1. “alfa-synuclein” should be “alpha-synuclein” or better yet “α-synuclein”

2. p. 5 -- BMRB ID 6869 appears to be a folded protein? The web page with this ID is associated with the “solution structure of the C-Terminal 14 kDa Domain of the tau subunit from Escherichia coli DNA Polymerase III”

3. line 228, the “Section” number is missing

4. line 207, I would think that a citation should go here to e.g. some of the chemical-shift based work by the Bax group on TALOS or CS-Rosetta.

5. why is the spread in Figure 4 between 98+% and 78+%? Is this due to an underlying AA compositional bias? Do all proteins all have the same AA distribution? It would be interesting to see a correlation between the accuracy and the AA diversity for each of the proteins in the training set. Having access to such information would enable the user to predict a priori how well the LDA approach will work on their system of interest based on the underlying AA composition. It may be important to re-train on a larger database in the future if certain AAs are poorly represented (e.g. aromatics and hydrophobics)

6. For table 1, I guess the acq parameters for directly-detected 1H (t3 or t4) are the same and hence omitted? And I guess that ni = ni + ni2 (+ni3)?

7. In a typical manual analysis of NMR data, it is usually trivial to identify and assign the XG/GX, XA/AX, XS/SX, and XT/TX spin systems, as the authors even write in their introduction. Thus, would it be possible to enable the user to manually “fix” specific assignments before running the calculation? You might envision that including e.g. the first 25% of assignments (G, A, S, T-containing spin systems) that are usually trivial to obtain might improve the LDA classification by having this starting point. I don’t know if it’s possible to do this though.

8. I commend the authors for making their data available on Zenodo. I downloaded the HNCO and HabCab(CO)NH spectra and looked at them in Sparky. I was surprised that the HNCO contained so much noise; was it properly reconstructed? The HabCab(CO)NH spectrum on the other hand was very clean with minimal artifacts.

**Have the authors made all data and (if applicable) computational code underlying the findings in their manuscript fully available?**

Reviewer #1: **No: **The software is not available. A Github page is announced in the header section, but no details are given in the paper.

Reviewer #2: **No: **the authors should provide the code for the LDA analysis

Reviewer #3: Yes

PLOS authors have the option to publish the peer review history of their article (what does this mean?). If published, this will include your full peer review and any attached files.

Reviewer #1: No

Reviewer #2: No

Reviewer #3: No
---

## [Decision Letter · Decision Letter 1]

20 Sep 2022

Dear Prof. Kazimierczuk,

We are pleased to inform you that your manuscript 'Linear discriminant analysis reveals hidden patterns in NMR chemical shifts of intrinsically disordered proteins' has been provisionally accepted for publication in PLOS Computational Biology.

Before your manuscript can be formally accepted you will need to address minor comments from one of the reviewers and complete some formatting changes, which you will receive in a follow up email. A member of our team will be in touch with a set of requests.

Best regards,

Anna R Panchenko

Academic Editor

PLOS Computational Biology

Arne Elofsson

Section Editor

PLOS Computational Biology

Reviewer's Responses to Questions

**Comments to the Authors:**

Reviewer #2: Overall, the authors have addressed many points raised. However, they have somehow failed to demonstrate the advantages of using their approach to assign IDPs. In theory, information about a.a. types could help, but the manuscript would significantly benefit from demonstration of such benefits for real systems. For example, the authors could incorporate information about residue types into AUTOASSIGN (or other NMR assignment software) and show the % of correctly assigned residues with and without residue type information obtained from their algorithm. From the examples in the manuscripts, I have an impression that requirements for data quality for LDA and for sequential methods are very similar, meaning that it's likely that LDA would provide no benefit/extra information if the data quality is limited (i.e., for the majority of IDPs).

**Have the authors made all data and (if applicable) computational code underlying the findings in their manuscript fully available?**

Reviewer #2: Yes

PLOS authors have the option to publish the peer review history of their article (what does this mean?). If published, this will include your full peer review and any attached files.

Reviewer #2: No

---

## [Editor Report · Acceptance letter]

3 Oct 2022

PCOMPBIOL-D-22-00827R1 

Linear discriminant analysis reveals hidden patterns in NMR chemical shifts of intrinsically disordered proteins

Dear Dr Kazimierczuk,

I am pleased to inform you that your manuscript has been formally accepted for publication in PLOS Computational Biology. Your manuscript is now with our production department and you will be notified of the publication date in due course.

With kind regards,

Agnes Pap
